# Adenomyosis in Pregnancy—Should It Be Managed in High-Risk Obstetric Units?

**DOI:** 10.3390/diagnostics13061184

**Published:** 2023-03-20

**Authors:** Rodrigo Orozco, José Carlos Vilches, Ignacio Brunel, Manuel Lozano, Gema Hernández, David Pérez-Del Rey, Laura Meloni, Juan Luis Alcázar

**Affiliations:** 1Department of Obstetrics and Gynecology, Hospital QuirónSalud, 29004 Malaga, Spain; 2Department of Computer Engineering, Universidad Politécnica, 28040 Madrid, Spain; 3Biomedical Informatics Group, Department of Artificial Intelligence, Computer Engineering, Universidad Politécnica, 28040 Madrid, Spain; 4TriNetX LLC, Cambridge, MA 02140, USA; 5Department of Obstetrics and Gynecology, Clínica Universidad de Navarra, 31008 Pamplona, Spain

**Keywords:** adenomyosis, diagnosis, obstetric complications, high-risk pregnancy

## Abstract

Background: Uterine adenomyosis is an increasingly frequent disorder. Our study aimed to demonstrate the presence of obstetric complications in the population affected by this condition to demonstrate the need for follow-up in high-risk obstetric units. Material and Methods: The data for the study were obtained from TriNetX, LLC, between 2010 and 2020. The outcomes analyzed were intrauterine growth restriction (IUGR), preterm delivery, cesarean delivery, hypertension, abruption placentae, and spontaneous abortion. Seven thousand six hundred and eight patients were included in the cohort of pregnant patients with adenomyosis, and 566,153 women in the cohort of pregnant patients without any history of endometriosis. Results: Upon calculating the total risk of presenting any of these problems during pregnancy, we obtained an OR = 1.521, implying that a pregnancy with adenomyosis was 52.1% more likely to present some complication. We found: IUGR OR = 1.257 (95% CI: 1.064–1.485) (*p* = 0.007); preterm delivery OR = 1.422 (95% CI: 1.264–1.600) (*p* = 0.0001); cesarean delivery OR = 1.099 (95% CI: 1.002–1.205) (*p* = 0.046); hypertensive disorders OR = 1.177 (95% CI: 1.076–1.288) (*p* = 0.0001); abruption placentae OR = 1.197 (95% CI: 1.008–1.422) (*p* = 0.040), and spontaneous abortion OR = 1.529 (95% CI: 1.360–1.718) (*p* = 0.0001). Conclusion: We conclude that the review carried out and the data we obtained on increased risk provide sufficient evidence to recommend that patients with adenomyosis should be managed in obstetric high-risk units.

## 1. Introduction

Adenomyosis is a benign pathology of the uterus, defined by the presence of endometrial glands and stroma within the myometrium, and until relatively recently, its diagnosis has only been considered possible with a histological examination of the uterus after hysterectomy [1]. The risk factors for adenomyosis include multiparity, cesarean delivery, early menarche, obesity, short cycles, curettage, and age of the woman. The pathophysiology of this disease is poorly understood. Two theories are currently accepted: the intussusception theory of the basal layer of the endometrium and the celomic metaplasia theory [2]. The first proposes that adenomyosis is the result of the invagination of the endometrial basal layer, due to the activation of a tissue injury and repair mechanism. An important role in this mechanism is played by local hyperestrogenism, as well as hyperpersitalsis of the uterine muscle. The second theory is based on the metaplasia of pluripotent embryonic Müllerian remnants or adult stem cells [2]. Some authors proposed a link between endometriosis and adenomyosis, considering the latter as a form of endometriosis, adding a third hypothesis for the pathophysiology of adenomyosis [3]. According to these authors, there are two forms of adenomyosis, i.e., external or extrinsic adenomyosis and internal or intrinsic adenomyosis. The intrinsic one affects the internal two-thirds of the myometrium and would be explained by the two theories previously exposed. The extrinsic form affects the outer third of the myometrium and could be explained by an implant theory.

The estimated incidence of adenomyosis is 290/100,000 woman-years using clinical or histological diagnosis. This incidence has not changed in recent years [4]. The most frequent age of diagnosis is 41–45 years [4]. If we consider the diagnosis of endometriosis together with adenomyosis, two age peaks are observed, one around 30 years, and the other around 50 years [5]. The prevalence of adenomyosis in hysterectomy specimens is estimated to be about 25% [1,6]. A recently published study using ultrasound criteria to diagnose this entity established that the prevalence was 21%, which is quite similar to the value reported in histological studies [7]. The prevalence of adenomyosis in patients operated on for endometriosis is 22% [8].

There are no pathognomonic symptoms associated with adenomyosis. The most common symptoms related to adenomyosis are pelvic pain (dysmenorrhea, dyspareunia, or chronic pelvic pain) occurring in 50–94% of the patients, abnormal uterine bleeding (27–65%), and infertility (30%). However, up to 30% of women with adenomyosis are asymptomatic [9].

Currently, ultrasound is considered the first-line imaging technique for the diagnosis of adenomyosis. Ultrasound findings in the uterus affected by adenomyosis have been studied for some time, and some of them are well known [10,11,12]. These include a globular uterus, myometrial asymmetry, fan shadows, myometrial cystic areas, echogenic islets, adenomyoma (poorly defined lesion of heterogeneous echogenicity located in the myometrial thickness) and increased focal myometrial vascularization (usually related to the presence of a focal lesion). More recently, three-dimensional ultrasound examination has also been reported as a useful technique, allowing a more detailed evaluation of the myo-endometrial junction, which is quite frequently altered in adenomyosis [13]. It usually appears thickened (>8 mm), irregular, or interrupted. The MUSA (Morphological Uterus Sonographic Assessment) group published a consensus in 2015 indicating the characteristics associated with adenomyosis that allow its differentiation from uterine fibroids [14].

Regarding the performance of transvaginal ultrasound for the diagnosis of adenomyosis, many studies were published since the early 1990s. Some meta-analyses that analyzed these studies reported that the sensitivity of ultrasound is 72–84%, and its specificity is 74–87% [15,16]. Several meta-analyses compared the diagnostic performance of transvaginal ultrasound with that of magnetic resonance imaging [17,18,19,20]. The conclusion of all these meta-analyses is that ultrasound offers similar diagnostic performance to MRI.

As stated above, adenomyosis might be related to infertility. However, considerable controversy exists over a possible link between adenomyosis and infertility, largely due to the limited size of the series studied [21,22]. Associations have also been postulated between uterine adenomyosis and complications during pregnancy, such as increased rates of spontaneous abortions and premature birth [23,24]. In spite of these findings, uterine adenomyosis is not traditionally considered a high-risk condition during pregnancy [25]. Our study aimed to investigate if this pathology may raise morbidity before and during pregnancy, which would make it an obstetrical high-risk condition to be preferentially managed in specialized high-risk maternal–fetal units.

## 2. Materials and Methods

The data for the study were obtained from TriNetX, LLC (“TriNetX”, Cambridge, MA, USA), a global federated health research network that provides access to EMRs from healthcare organizations (“HCOs”) all over the world. TriNetX provides access to data containing diagnoses, procedures, medications, laboratory values, and genomic information from approximately 250 M patients from over 130 HCOs. Data collection, processing, and transmission were performed in compliance with all data protection laws applicable to the contributing HCOs, including the EU Data Protection Law Regulation 2016/679, the General Data Protection Regulation on the protection of natural persons with regard to the processing of personal data and the Health Insurance Portability and Accountability Act (“HIPAA”), and the US federal law, which protects the privacy and security of healthcare data. The TriNetX Global Collaborative Network (TGCN) is a distributed network, and analytics are performed on anonymized or pseudonymized/de-identified (per HIPAA) data housed at the HCOs, with only aggregate results being returned to the TriNetX platform. Individual personal data do not leave the HCO. TriNetX is ISO 27001:2013-certified and maintains a robust IT security program that protects both personal data and healthcare data. The presented study included data from subjects from the TGCN that had a medical record of pregnancy between 2010 and 2020. The cohort was queried in TriNetX on 12 July 2022.

### 2.1. Cohort and Outcome Definition

The patients included in this study were female with a record of the ICD-10-CM code Z33, *Pregnant state*, between 1 January 2010 and 1 January 2021. The patients must have been 18 years old or older at the time of pregnancy. To ensure that the study only included patients with previous medical history at the corresponding HCO, the patients were required to have a visit at least 6 months before the first pregnancy record. Patients with a record of the ICD-10-CM N80.0, *Endometriosis of uterus*, were included in the adenomyosis cohort. These were compared to the cohort of patients without any record of endometriosis, i.e., no record of N80, *Endometriosis*.

The exclusion criteria for the study group were: patients under 18 years of age at the time of pregnancy, patients with no prior history of at least 6 months in any of the network centers, patients with endometriosis but without adenomyosis. The following exclusion criteria were established for the control group: patients under 18 years of age at the time of pregnancy, patients with no prior history of at least 6 months in any of the network centers, patients with any type of endometriosis.

The outcomes analyzed in this study were intrauterine growth restriction (IUGR), preterm delivery, cesarean delivery, hypertension, abruption placentae, and spontaneous abortion. IUGR was defined with the ICD-10 codes Z36.4, *Encounter for antenatal screening for fetal growth retardation*, and O36.59, *Maternal care for other known or suspected poor fetal growth*. Preterm delivery was captured with the ICD-10-CM code O60, *Preterm labor*. The cesarean delivery outcome was defined as having a record of any of the following: ICD-10-PCS 10D00Z0, 10D00Z1, 10D00Z2, ICD-10-CM code O82, *Encounter for cesarean delivery without indication*, and SNOMED-CT code 11466000, *Cesarean section*. The definition of hypertension included the ICD-10-CM codes O13, *Gestational [pregnancy-induced] hypertension without significant proteinuria*, O14, *Pre-eclampsia*, O15, *Eclampsia*, and O16, *Unspecified maternal hypertension*. The condition of abruption placentae was defined according to the ICD-10-CM concepts O43, *Placental disorders*, and O45, *Premature separation of placenta [abruption placentae]*. Finally, spontaneous abortion was defined according to the ICD-10-CM code O03, *Spontaneous abortion*. The age at the time pregnancy was considered a possible confounding variable to be balanced using propensity score matching.

The data available in TriNetX came mostly from structured EHR sources of the HCOs from TGCN. For the study, only data attributed to the mother were used. Data attributed to the newborn were not used, as these are considered highly sensitive data by the TriNetX Data Privacy principles.

### 2.2. Patients

A total of 7608 patients (97% from the USA) coming from 86 HCOs of the network were included in the cohort of pregnant patients with adenomyosis. A total of 566,153 patients from the same 86 HCOs of the network were included in the cohort of pregnant patients without any history of endometriosis. The patients in the adenomyosis cohort were significantly older (Table 1).

Most women in both cohorts were from the United States of America (USA) (Table 2).

### 2.3. Statistical Analysis

All statistical analyses were run on the TriNetX Platform. TriNetX is a custom-developed solution built using Java for the backend and some R and Python statistical analysis libraries, such as Hmisc, lifelines, matplolib, numpy, pandas, and scipy.

For each of the outcomes analyzed in this study, several metrics of association between the cohorts were calculated. The risk difference between the cohorts was assessed with a Z-test, providing a 95% confidence interval. The risk ratio was also calculated with a 95% confidence interval. The patients were censored when the last fact in a patient’s records appeared within the time window of the analysis. The time window used in this analysis was from the same day of the index event, the first pregnancy record, until one year later. The age at the time of the first pregnancy record was balanced by 1:1 propensity score matching. The propensity scores were calculated using logistic regression of the well-tested, standard Python package scikit-learn.

## 3. Results

Overall, the total risk of presenting any of the complications evaluated during pregnancy was higher in women with adenomyosis (OR: 1.521), implying that a pregnancy with adenomyosis is 52.1% more likely to present some complication.

### 3.1. Intrauterine Growth Restriction

The incidence of intrauterine growth restriction was 4.3% in the adenomyosis cohort vs. 3.4% in the cohort of patients without endometriosis. The OR was 1.257 (95% CI 1.064–1.485), with a *p* value = 0.007. This implies a 25% greater risk of presenting intrauterine growth in these patients.

### 3.2. Preterm Delivery

The incidence of preterm delivery was 9.4% in the patient cohort with adenomyosis vs. 6% in the cohort without endometriosis. The OR was 1.422 (95% CI 1.264–1.600), with a *p* value = 0.0001, corresponding to a 42% increase in the risk of presenting preterm delivery in these patients.

### 3.3. Cesarean Delivery

The incidence of a cesarean delivery outcome was 14.3% in the cohort of patients with adenomyosis vs. 13.2% in the cohort without endometriosis. The OR was 1.099 (95% CI 1.002–1.205), with a *p* value = 0.046, corresponding to an increased risk of cesarean delivery outcome of 9.9% in these patients.

### 3.4. Maternal Hypertension

Some type of hypertensive pathology was detected during pregnancy in 15.7% of the patient cohort with adenomyosis versus 13.7% of the patient group without endometriosis. The OR was 1.177 (95% CI 1.076–1.288), with a *p* value = 0.0001. These patients therefore had a 17.7% greater risk of presenting some hypertensive condition during pregnancy.

### 3.5. Abruption Placentae

Abruption placentae occurred in 3.9% of the patient cohort with adenomyosis vs. 3.3% of the patient cohort without endometriosis, with an OR of 1.197 (95% CI (1.008–1.422) and a *p* value = 0.040. Therefore, the risk of presenting abruption placentae increased by 19% in these patients.

### 3.6. Spontaneous Abortion

The incidence of spontaneous abortion reached 10% in the adenomyosis cohort vs. 6.8% in the patients without endometriosis. The OR was 1.529 (95% CI 1.360–1.718), with a *p* value = 0.0001, representing an increased risk of 52.9% of spontaneous abortion in these patients.

These findings are summarized in Table 3.

## 4. Discussion

Numerous studies attempted to demonstrate that adenomyosis is an independent risk factor for obstetric complications. It is important to clarify this situation, owing to the need to follow up patients with this condition in high-risk obstetrics units. However, the main problem with these studies is their small sample size, which reduces the power of the studies, and observe odd ratios for most obstetrical complications with broad confidence intervals, which reflects the heterogeneity of the sample and might lead to conclusions with low precision and reliability. Even in studies on national cohorts, such as the one conducted by Yamaguchi et al. on a Japanese cohort during 2011 and 2014, the number of pregnancies with adenomyosis was just 314 [26]. Additionally, the infertility associated with adenomyosis and the need to use assisted reproduction techniques may contribute to the difficulty of recruiting a suitable cohort of these patients.

For this reason, some researchers tried to summarize the current evidence on the risks associated with adenomyosis during pregnancy in several literature reviews, such as the ones carried out by Harada et al. [27], Buggio et al. [28], and Wendel et al. [29]. However, these papers just reviewed the literature and speculated about the potential mechanisms that explain why adenomyosis can produce poorer obstetrical outcomes [27] and highlighted several methodological drawbacks of the studies reported, such as poor design, small sample size, high qualitative heterogeneity, potential misclassification bias, and co-morbid conditions; this prevents drawing definitive conclusions on the strength of the observed associations and on the magnitude of the treatment effects [28]. Another common problem in the recruitment of patients with adenomyosis lies in the confirmation of the diagnosis. Although, classically, diagnoses have been based on a pathological confirmation, thanks to advances in ultrasound and MRI, very accurate diagnoses can now be made without the need for surgery [19,20].

Some systematic reviews and meta-analyses analyzing the potential impact of adenomyosis in late obstetrical complications have been reported. Horton et al. reported in 2019 on five studies, comprising a total of 330 women with an ultrasound diagnosis of adenomyosis and 9384 controls [24]. They observed an increased risk for spontaneous abortion (OR: 3.49, 95% CI: 1.41–8.65), pre-eclampsia (OR: 7.87, 95% CI: 1.26–49.20), small for gestational age (OR: 3.90, 95% CI: 2.10–7.25), preterm delivery (OR: 2.74, 95% CI: 1.89–3.97), and cesarean section (OR: 2.62, 95% CI: 1.00–6.89). Nirgianakis et al. reported more recently another meta-analysis on seventeen studies comprising 1115 women diagnosed as having adenomyosis and 96,225 controls [29]. These authors observed an increased risk for spontaneous abortion (OR: 2.50, 95% CI: 1.26–4.95), pre-eclampsia (OR: 4.32, 95% CI: 1.68–11.09), small for gestational age (OR: 2.10, 95% CI: 1.17–3.77), preterm delivery (OR: 2.63, 95% CI: 1.96–3.54), and cesarean section (OR: 4.44, 95% CI: 2.64–7.46). Certainly, most studies involved women who underwent an IVF treatment and few women with spontaneous conception.

Our results are in agreement with the results reported in these meta-analyses. We observed poorer obstetrical outcomes in women with adenomyosis. However, we consider the sample size is the main strength of our study. A cohort the size of ours, with 7608 pregnant women with adenomyosis, has not been reported previously in the literature. Moreover, here we compared this cohort with a large control cohort of 566,131 patients, which permitted us to perform propensity score matching of our patients in a ratio of 1:1, to reduce bias or the influence of differences other than the presence of adenomyosis. Ours is also the first study to analyze so many obstetric complications in a single cohort. The most frequent complications found in our study corresponded to spontaneous abortion, with an OR of 1.529 (an increased risk of 52% compared with the normal population), premature birth, with an OR of 1.422 (corresponding to an increased risk of 42% higher than in the normal population), and restricted intrauterine growth, with an OR of 1.257 (reflecting a risk 25% higher than in the normal population). We also observed a higher incidence of severe complications such as hypertension, OR 1.17, or abruption placentae, OR 1.19, a complication accounting for 9.2% of perinatal mortality [30]. Finally, the rate of cesarean delivery was also slightly higher in our series, with an OR of 1.09. Another strength of our study is that we obtained data from a large database, with data from several geographical regions, which can make our results generalizable.

However, our study has some limitations. The main limitation is that our study was a retrospective study analyzing data from a database. Therefore, we could not avoid some kind of bias. Nonetheless, in our opinion, the most important piece of data in our study is the sum of the total risk of presenting some problem during pregnancy, which reached an odds ratio of 1.521 in our patients. This implies that patients with adenomyosis have a 52.1% higher risk of suffering some complication during their pregnancy than patients without this condition.

The underlying mechanisms through which adenomyosis causes poor obstetrical outcomes have not yet been elucidated. There are some proposed explanations such as immunological changes, inflammation, increased uterine pressure, increased prostaglandin expression, increased myometrial wall thickness, and abnormal trophoblastic invasion and placentation due to an abnormal myometrial–endometrial junctional zone [27]. For example, a eutopic endometrium in women with adenomyosis shows aberrant immunological changes contributing to implantation failure, which could explain the high abortion rate [27]. Some studies showed that the rate of uterine infection in patients with diffuse-type adenomyosis was higher than that in patients with focal-type adenomyosis [31,32]. Additionally, it is also known that sterile inflammation can induce a common inflammatory pathway leading to labor and contribute to the onset of parturition and preterm birth [33]. The pathogenic processes associated with microbial or sterile inflammations can induce a common pathway resulting in preterm birth. Furthermore, some studies reported an increased intrauterine pressure in patients with adenomyosis, which may also lead to preterm labor and birth [27]. A recent study assessing myometrial thickness by ultrasonography showed that the uterine wall was thicker and less stretched during the second trimester of pregnancy in women who had a preterm delivery [34]. All these mechanisms could explain the higher rate of preterm delivery in women with adenomyosis.

Regarding the complications intrauterine growth restriction or small for gestational age, several mechanisms have been advocated, among which placental dysfunction is the main one [27]. The main hypothesis is that placental development close to the adenomyotic myometrium might lead to defective placentation with a smaller placenta and poorer placental vascularization and blood flow [27]. The placental abruption higher risk in the presence of adenomyosis could be explained also by an abnormal placentation process that may lead to vascular disruption and bleeding, resulting in thrombin production that may play a role in placental detachment [27]. It is interesting to note that an increase in the pulsatility index of the uterine arteries in the third trimester in women with endometriosis has been recently described [35]. This indicates that abnormal placentation occurs in women with this disease. All mechanisms above-described may also explain the higher risk for developing pre-eclampsia, observed in women with adenomyosis.

It has been speculated that different subtypes of adenomyosis (focal or diffuse) might cause different obstetrical complications. It has been shown that women diagnosed with diffuse adenomyosis had a higher risk of spontaneous abortion, pre-eclampsia, and preterm delivery than women diagnosed with the focal type [27]. The inner uterine layer is an integral component of normal decidualization and placentation. In particular, the integrity of the junctional zone or endometrial–myometrial border layer may significantly affect the establishment of pregnancy, and alterations of this inner layer may have negative consequences for pregnancy outcomes. However, prospective studies, which explore the divergent obstetric complications according to the subtype of adenomyosis, are needed to expand these findings.

## 5. Conclusions

There is an increased risk of obstetric complications associated with uterine adenomyosis. The pathophysiological explanation of adverse pregnancy outcomes in women with adenomyosis may include many mechanisms, and the complex biochemical pathways remain unclear. Complications might be caused by the thickening of the myometrium, abnormal uterine distensibility, and deformation of the uterine cavity.

We consider that our data regarding the increased risk of obstetric complications provide sufficient evidence to recommend that patients with adenomyosis should be managed in special high-risk obstetric units.

## Figures and Tables

**Table 1 diagnostics-13-01184-t001:** Characteristics of the participants with pregnancy with adenomyosis (cohort 1) and pregnancy without endometriosis (cohort 2), before propensity score matching.

Cohort	Age at Index Mean ± SD	N of Patients	% of Cohort	*p*-Value
1	37.5 ± 11.5	7607	100%	<0.001
2	34.0 ± 14.1	566,131	100%	

**Table 2 diagnostics-13-01184-t002:** Geographical distribution of the patient cohorts.

	Pregnancy without Endometriosis	Pregnancy with Adenomyosis
USA Northeast	33%	33%
USA Midwest	11%	14%
USA South	47%	46%
USA West	6%	4%
Non-USA	3%	3%

**Table 3 diagnostics-13-01184-t003:** Summary of the outcomes with propensity score matching.

	Adenomyosis	No Endometriosis	OR	Or 95% CI	*p* Value
Intrauterine Growth restriction	324 (4.3%)	260 (3.4%)	1.257	(1.064, 1.485)	0.007
Preterm Delivery	716 (9.4%)	518 (6.8%)	1.422	(1.264, 1.600)	0.0001
Cesarean Delivery	1091 (14.3%)	1006 (13.2%)	1.099	(1.002, 1.205)	0.046
Hypertension	1194 (15.7%)	1039 (13.7%)	1.177	(1.076, 1.288)	0.0001
Abruption Placentae	295 (3.9%)	248 (3.3%)	1.197	(1.008, 1.422	0.040
Spontaneous Abortion	760 (10.0%)	515 (6.8%)	1.529	(1.360, 1.718)	0.0001

## Data Availability

Not applicable.

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
