# Peer review of "Adenomyosis in Pregnancy—Should It Be Managed in High-Risk Obstetric Units?"

_diagnostics, 2023, doi:10.3390/diagnostics13061184_

Round 1

Reviewer 1 Report

1. Reconsider the introduction: Risk factors for adenomyosis include .......termination of pregnancy ? (more details); An important role in this mechanism is played by local hyperstronism?( I don’t understand); the introduction is not concise.

2. Influence of assisted reproductive technology on pregnancy outcomes in women with adenomyosis in in this study ?

3. Any correlations between age of pregnant women with adenomyosis and pregnancy outcomes ?

4. Why didn't you include preterm premature rupture of membranes (PPROM) in the analysis of pregnancy outcomes ?

5. Can you specify if the patients included in the study had a history of previous uterine surgery or comorbidities ?

6. The exclusion criteria are not clearly specified (my opinion)

Author Response

Dear Reviewer

Thanks for your comments. We appreciate them

These are our responses.

  1. Reconsider the introduction: Risk factors for adenomyosis include .......termination of pregnancy? (more details); An important role in this mechanism is played by local hyperstronism?( I don’t understand); the introduction is not concise.

We have modified the introduction to make it more concise and explain the points that generated more doubts

Deleted termination of pregnancy, since the risk factor is termination by cesarean section.

Change hyperstronism by hyperestrogenism

  1. Influence of assisted reproductive technology on pregnancy outcomes in women with adenomyosis in in this study ?

We consider that the influence of assisted reproduction is ruled out since the comparison cohort had the same rate of assisted reproduction as the patients with adenomyosis. Therefore, a propensity score was performed to reduce possible biases between the group of patients with and without adenomyosis.

3.Any correlations between age of pregnant women with adenomyosis and pregnancy outcomes ?

Although it is true that there is a difference in the average age of the two groups, we consider it to be small. However, precisely for this reason we have carried out the propensity score whose idea is to homogenize the two groups as much as possible based on a characteristic in which they differ. We consider that by screening 7607 patients versus 566131 patients we have limited the influence of this difference as much as possible.

  1. Why didn't you include preterm premature rupture of membranes (PPROM) in the analysis of pregnancy outcomes ?

We appreciate your suggestion. Of course it is interesting to include as many variables as possible. However, premature rupture of membranes is not per se a complication in the course of pregnancy/delivery and we have preferred to include preterm delivery in the study.

  1. Can you specify if the patients included in the study had a history of previous uterine surgery or comorbidities ?

Trinetx database with which the study was carried out allows us to know in absolute numbers the patients studied according to the variables analyzed.

To avoid bias and, above all, to reduce the risk of bias, multiple statistical studies have been applied to homogenize the comparison and study groups as much as possible.

It would be interesting for a future article to evaluate the reviewer's proposal regarding the surgical history.

  1. The exclusion criteria are not clearly specified (my opinion)

It is true that the exclusion criteria are not clearly defined. We added these criteria to the manuscript.

Reviewer 2 Report

The authors explore the possible relation between adenomyosis and adverse pregnancy outcomes by examining diagnostic codes from large routinely collected database. The data spans 10 years 2010-20.

Overall, the data presented is interesting, but there is a significant limitation because 1) there is a significant difference in Age between the compared groups 2) the full characteristics of the two groups are not provided 3) the method by which adenomyosis is diagnosed or endometriosis is ruled out is not provided.

The introduction: Contains a synopsis about adenomyosis, its diagnosis and theories. But this has no direct focus on the article and is mostly redundant. The reader is most likely to be familiar with the data provided - furthermore, there is no indication as to whether or to what extent US, MRI or histology were used in the diagnosis. There is no indication of the temporal relationship between the diagnosis and the occurrence of pregnancy or its complications. There is no indication as to other factors such as infertility, concomitant uterine or systemic pathology.

Line 46: Regarding ... either: please revise. 

Line 47 and 48: accepted: it is rather debated. What is meant by established?

Line 52: establish: please revise.

Line 58: and would rather be explained.....theory: please revise.

line 69: you mean Pathognomonic.

line 102: you mean preferably? Are there (or many of these) patient not likey to be seen in high risk set-ups anyway because of other comorbidities?

M&M: how is the database validated to be accurate?

line 246: recur?

Discussion:

A large part of the discussion is consumed by speculation about possible mechanisms - when more attention should be focussed on assessment of existing literature and a critical review of this study and its limitations and reflection of what this study adds to knowledge.

Conclusion:

this also contains an entry about pathophysiology that is merely a speculation and is not rooted in the study presented.

Author Response

Dear reviewer

Thanks for your comments

These are our responses

The authors explore the possible relation between adenomyosis and adverse pregnancy outcomes by examining diagnostic codes from large routinely collected database. The data spans 10 years 2010-20.

Overall, the data presented is interesting, but there is a significant limitation because 1) there is a significant difference in Age between the compared groups 2) the full characteristics of the two groups are not provided 3) the method by which adenomyosis is diagnosed or endometriosis is ruled out is not provided.

The introduction: Contains a synopsis about adenomyosis, its diagnosis and theories. But this has no direct focus on the article and is mostly redundant. The reader is most likely to be familiar with the data provided - furthermore, there is no indication as to whether or to what extent US, MRI or histology were used in the diagnosis. There is no indication of the temporal relationship between the diagnosis and the occurrence of pregnancy or its complications. There is no indication as to other factors such as infertility, concomitant uterine or systemic pathology.

Line 46: Regarding ... either: please revise. 

Line 47 and 48: accepted: it is rather debated. What is meant by established?

Line 52: establish: please revise.

Line 58: and would rather be explained.....theory: please revise.

line 69: you mean Pathognomonic.

line 102: you mean preferably? Are there (or many of these) patient not likey to be seen in high risk set-ups anyway because of other comorbidities?

M&M: how is the database validated to be accurate?

line 246: recur?

Discussion:

A large part of the discussion is consumed by speculation about possible mechanisms - when more attention should be focussed on assessment of existing literature and a critical review of this study and its limitations and reflection of what this study adds to knowledge.

Conclusion:

this also contains an entry about pathophysiology that is merely a speculation and is not rooted in the study presented.

Answer

Thank you very much for your comments. We consider that they allow us to improve this article and obtain the best version of it. Regarding your comments:

1)Although it is true that there is a difference in the average age of the two groups, we consider it to be small. However, precisely for this reason we have carried out the propensity score whose idea is to homogenize the two groups as much as possible based on a characteristic in which they differ. We consider that by screening 7607 patients versus 566131 patients we have limited the influence of this difference as much as possible.

2) We add the following table for the reviewer's study on some additional data related to the sample

pregnant with adenomyosis

pregnant without endometriosis

Demographics

Mean ± SD

Min

Max

Patients

% of Cohort

Mean ± SD

Min

Max

Patients

% of Cohort

AI

Age at Index

37.5 ± 11.5

18

87

7608

100%

34 ± 14.1

18

89

566153

100%

2106-3

White

-

-

-

4718

62%

-

-

-

356621

63%

2054-5

Black or African American

-

-

-

1886

25%

119216

21%

2131-1

Unknown Race

-

-

-

799

11%

-

-

-

72312

13%

2028-9

Asian

-

-

-

170

2%

-

-

-

14938

3%

3) The diagnosis of adenomyosis was made by MRI or high-resolution transvaginal ultrasound in a referral center. We add this point to the manuscript

4) We have modified the introduction to make it more concise and explain the points that generated more doubts

5) line 46 fixed

6) Although there is no theory considered as the main theory or gold standard, we do consider that these two theories are the most widely accepted in the literature. Changed  established by proposes.

7) Line 52 changed

8) Line 58 changed

9) Line 69 fixed

10) Normally, patients who only present adenomyosis do not have to present other comorbidities that justify their gestational follow-up in a high-risk unit. In any case, the objective of this article is to justify that regardless of the other characteristics that the adenomyosis itself presents, it is a compelling reason for them to be more monitored.

11) Trinetx is an international database of recognized prestige and with more than 300 articles indexed in Pubmed. We consider that the validity of this database is more than demonstrated. The procedure for filtering cases on this occasion is described in material and methods

12) Line 246 changed

13) We update the discussion

14) Conclusion: We change the structure as well as the order of the structure to make it internally consistent.

Round 2

Reviewer 2 Report

Thank you - the authors have addressed the points. There remains weakness in design, because of the limitations of diagnosis of the conditions. The differences in adverse outcomes may still be related to differences in age and in other population factors (and these have not been fully explored).

Author Response

1